# Quality Estimation & Interpretability for Code Translation

Mayank Agarwal [* 1]  Kartik Talamadupula [* 1]  Stephanie Houde [1]  Fernando Martinez [2]  Michael Muller [1]
John Richards [1]  Steven Ross [1]  Justin D. Weisz [1]

## Abstract

Recently, the automated translation of source code from one programming language to another by using automatic approaches inspired by Neural Machine Translation (NMT) methods for natural languages has come under study. However, such approaches suffer from the same problem as previous NMT approaches on natural languages, viz. the lack of an ability to estimate and evaluate the quality of the translations; and consequently ascribe some measure of interpretability to the model's choices. In this paper, we attempt to estimate the quality of source code translations built on top of the TransCoder (Roziere et al., 2020) model. We consider the code translation task as an analog of machine translation (MT) for natural languages, with some added caveats. We present our main motivation from a user study built around code translation; and present a technique that correlates the confidences generated by that model to lint errors in the translated code. We conclude with some observations on these correlations, and some ideas for future work.

## 1. Introduction

The emergence of large-scale, automatically-trained models in Natural Language Processing (NLP) has enabled many interesting and novel applications. Of particular interest to us in this work is the area of *Machine Translation* (MT), where translations from one (natural) language to the other are produced automatically. Neural techniques applied to the machine translation task (Garg & Agarwal, 2018) have previously produced state-of-the-art (SoTA) results. As machine translation systems improved the fluency of translations between (human) natural languages, a pressing imperative that came to the fore was *Quality Estimation (QE)* of these MT systems in an automated manner. While most previous work in this area (Kim et al., 2017; Wang et al., 2018; Kepler et al., 2019) models QE as a supervised machine learning

task, the work of Fomicheva et al. (2020) treats the NMT model as a glassbox, and proposes various unsupervised QE measures that correlate to various degrees of human judgment of the translations.

Code translation from one programming language to another can be viewed as a special case of machine translation on natural languages, albeit with rigorous grammar rules and a stricter notion of correctness as compared to translation between natural languages (Allamanis et al., 2018). Several studies have investigated the application of machine translation to the problem of code translation (Nguyen et al., 2013; Karaivanov et al., 2014; Oda et al., 2015); and more recently, deep neural networks (Chen et al., 2018) and unsupervised neural machine translation (NMT) techniques have been applied (Roziere et al., 2020) to this task as well.

Inspired by this progress, we present a study of translations from the TransCoder (Roziere et al., 2020) system. We extract the confidences produced by the system while translating $400$ source code programs from `Java` to `Python`; and seek to correlate these confidences to lint errors in the automatically-produced `Python` code snippets. In doing this, we rely on insights gleaned from a user study of software engineering professionals engaged in an application modernization task at a major information technology company. We intend for this report to: (i) spur further work on creating and studying metrics for the automatic estimation of code translation quality; and (ii) provide interpretations to these metrics that can be understood in the context of representations that software professionals are familiar with.

## 2. User Study

As the first step towards measuring the quality of translations produced by TransCoder, we conducted an interview study of 11 software engineering professionals – the most likely target group for the source code translation use case. This user study – conducted at a major multinational information technology company – provided some insights and motivation for our work, which we detail in this section.

The goal of our user study was to *learn how programmers respond to a utility that helps them translate code from* `Java` *to* `Python`, *potentially with imperfections* [1] . In addition,

---

[*]Equal contribution  [1]IBM Research AI, USA [2]IBM Argentina, Argentina. Correspondence to: Mayank Agarwal <mayank.agarwal@ibm.com>, Kartik Talamadupula <krtalamad@us.ibm.com>.

---

[1]This kind of task has practical value to organizations as they

we examined whether and how indicators of translator confidence and pop-up menus showing different translation options were perceived as useful. The recruiting profile included the requirement that participants be programmers who were familiar with both Java and Python (our languages of focus in this paper). We spent hour-long sessions with the 11 software engineers, showing them design prototypes and getting feedback – one configuration is shown in Figure 3 (Appendix). Tokens that the TransCoder model is less confident about are highlighted in red.

## 2.1. Preliminary Insights

The user study produced numerous qualitative insights[2]. In this section, we focus only on those aspects that related to participants' understanding – or lack thereof – of what the translator model was doing; and their suggestions on improving the interpretability of the translation process.

### 2.1.1. SYNTAX & STYLING

> "*In groups that try to adhere to the style guides produced by language creators...better to train the AI with code that adhered to a style guide than not...reading other people's code easier if everyone adheres to the same style.*" - P5

One of the main high-level insights – exemplified by the quote above from a participant – concerned the styling and syntax issues that are inherent in code. When code is being translated into a new language, it is not sufficient merely to produce something that can execute: the translation must also adhere to the conventions – written and unwritten – of the target language (Allamanis et al., 2014). This importance is reflected in various lint-like tools that are used to enforce constraints on and optimize code. This issue also has a distinct connection to *neural style transfer* (Jing et al., 2020); where the *style* of the target language can be seen as conventions that surround syntax and styling issues.

### 2.1.2. CONFIDENCE OF THE MODEL

> "*That just confuses me because this line of code is syntactically correct. I would write that line of code with 100% confidence. I'm confused about why the translator would be confused about that assignment.*" - P0

Another common theme was that participants were often left confused by the model's confidence – or more often, the lack thereof – in specific tokens in the translation. This confusion is best exemplified in the quote above, where participant P0 confirmed that the translated line of code was syntactically correct; yet noted with surprise the model's lack of confidence in the translation. Similar comments on

modernize legacy applications (Khadka et al., 2012).

___

[2]We are in the process of publishing the larger user study and will share a reference to it in a future version of this manuscript.

the punctuation tokens in translations came from participant P1: "*Why is the left bracket after the return... why is that not so confident... it's not even an identifier... the double quote in that string there, that's a funny one too.*" Participant P3 expected that the "*translator would be able to define this def function signature line with the function name correct*", but expressed confusion when the translator wasn't "*as confident as it could be*". Participant P5 questioned: "*Why is it highlighting so many of these comments?*" in reference to the model's uncertainly in code comments.

All of these quotes point to a central shortcoming with models like TransCoder: human users do not have a very good understanding of how the model performs the translation; and hence have very little idea about when and why it has confidence (or not) in a particular token. An important reason for this shortcoming is the difference in the way that humans generate translations versus neural models.

### 2.1.3. MAPPING TO INTERMEDIATE REPRESENTATIONS

The fundamental observation above transfers from the realm of MT for natural languages to the code domain. Toral et al. (2018) have examined this issue – and its implications for the evaluation of human parity in MT systems – in detail for the natural languages Chinese (ZH) and English (EN). Our user study gives us some preliminary indications that something similar happens when it comes to source code; where a human programmer who is proficient in Java and Python would translate (and check translations) different from the way an NMT model (like TransCoder) might do so. Specifically, human translations tend to: (i) consider a much larger and more global context when evaluating and producing translations; and (ii) map both the source and target (if available) material on to some common intermediate representation (for e.g., the concept of a *loop*). NMT models, on the other hand, often maximize the probability of the next token given the evidence of some restricted set of tokens in the neighborhood of that next token; and show no evidence of being able to map on to any representation that is in common with humans (programmers).

This insight motivates our study of the interpretability of the output of NMT models in this paper. Specifically, in the following sections, we seek to use lint errors as an intermediate representation that is already familiar to human users; and try to correlate the NMT model's confidence values with errors and warnings produced by the linter.

## 3. Experiments

We set up our experiments to translate a complete source code program (as against a function level translation) to help us understand the effect of auxiliary code blocks – such as import statements, class definitions, and multiple function definitions – on the translation quality.

| Code | E0602 | E0001 | E0102 | E1101 | E1102 | E0601 | R0903 | R1705 | R0205 | R1716 |
|---|---|---|---|---|---|---|---|---|---|---|
| $n$ | 218 | 74 | 42 | 34 | 24 | 18 | 70 | 67 | 55 | 14 |
| $r_{pb}^{joint}$ | 0.112 | 0.053 | 0.089 | 0.044 | 0.007 | 0.013 | 0.053 | -0.032 | 0.058 | -0.005 |
| $r_{pb}^{min}$ | 0.100 | 0.052 | 0.091 | 0.038 | 0.005 | 0.013 | 0.043 | -0.026 | 0.047 | -0.005 |

| Code | W0612 | W0622 | W0613 | W0621 | W0611 | C0103 | C0301 | C0200 | C0304 | C0325 |
|---|---|---|---|---|---|---|---|---|---|---|
| $n$ | 128 | 104 | 58 | 14 | 12 | 277 | 70 | 35 | 20 | 15 |
| $r_{pb}^{joint}$ | 0.069 | -0.001 | 0.073 | 0.012 | 0.056 | 0.083 | 0.085 | 0.006 | 0.012 | 0.006 |
| $r_{pb}^{min}$ | 0.064 | 0.001 | 0.061 | 0.013 | 0.055 | 0.071 | 0.077 | 0.004 | 0.014 | 0.009 |

*Table 1.* PBCC values and sample sizes for Pylint error (**E**), refactor (**R**), warning (**W**), and convention (**C**) messages. "Code" indicates the Pylint error code (refer Appendix A.2 for details) for which the correlation metric is computed. $n$ indicates the number of translations (out of 400) in which the particular error code is observed, and $r_{pb}^{joint}$ and $r_{pb}^{min}$ correspond to PBCC metrics using $\Upsilon_{\text{joint}}$ and $\Upsilon_{\text{min}}$ as token uncertainty values.

### 3.1. Method

One of the primary aims of this work is to show that there is very little correlation between the confidence measures output by a code translation model (like TransCoder) and traditional methods used by software engineers to check their code (like lint). Since we are interested in evaluating this correlation, we must first determine the two variables being correlated. The first such variable is continuous, and is simply the output from the TransCoder model for each token in the translated source code: $p(y_t \mid y_{<t}, x, \theta)$. The second variable is discrete/categorical; and takes the form of the error category that is flagged for a given line by running the translated source code through a linter.

#### 3.1.1. DATA

We utilize 400 common algorithmic implementations in `Java` downloaded from a popular Github repository (Pradhan & Pop, 2017), and produce a `Python3` translation for each of these instances using a pre-trained TransCoder model with a beam size of 5. For each translation, we also record the output probabilities associated with each token.

#### 3.1.2. LINT ERRORS

We ran each of the 400 `Python3` translations produced by TransCoder through the static code analysis tool `Pylint` to look for programming errors, coding standard violations, and simple refactoring suggestions. We execute `Pylint` to validate for all but three of the 311 violations included in the Pylint default configuration . Please refer to Appendix A.2 for `Pylint` related details. Some of these validations are checks for proper syntax, package import errors, undefined variable access or usage before assignment, redefining `Python` builtin functions or variable; among others.

### 3.2. Calculating Correlation

For our correlation analysis, we use the *Point Biserial Correlation Coefficient* (PBCC) (Tate, 1954), and its implementation in `SciPy` (Virtanen et al., 2020). The PBCC metric is typically used in scenarios where one variable is

dichotomous – that is, it can be partitioned into two mutually exclusive sets that are jointly exhaustive – and the other variable is continuous. The biserial correlation is when the variable is artificially dichotomized over some underlying continuity. In the case of our investigation, the dichotomous variable is whether a particular line of source code throws a linter error (of a specific category) or not. The continuous variable is taken to be an estimate of the model's uncertainty for the corresponding source code line. We consider two specific uncertainty metrics: $\Upsilon_{\text{joint}}$ computing the uncertainty based on the joint distribution over the $T$ tokens in the line; and $\Upsilon_{\text{min}}$ using the minimum token confidence value as an estimate of the line uncertainty:

$$\Upsilon_{\text{joint}} = 1 - \prod_{t=1}^{T} p(y_t | y_{<t}, x, \theta) \qquad (1)$$

$$\Upsilon_{\text{min}} = 1 - \min_{\forall t \in \{1, \cdots, T\}} p(y_t | y_{<t}, x, \theta) \qquad (2)$$

## 4. Results & Discussion

We focus on 3 main results: (1) a translation error analysis that offers a profile of the kinds of lint errors that occur in code translated by TransCoder; (2) a quantitative study of the correlation (or lack thereof) between model confidence values and lint errors; and (3) a qualitative example that drills deeper into one specific translation, and the correlations between TransCoder's confidence values and the lint errors for that translation.

### 4.1. Translation Error Analysis

To understand how TransCoder handles the syntactic differences between two programming languages – `Java` and `Python3` in our case – we identify the different kinds of lint violations that occur in the translated code. Figure 1 shows the top lint violations (out of the 66 observed) and the frequency with which they occur in the generated translations. The most frequently-occurring violation was `invalid-name`, where the model did not comply with the naming convention of a function or a variable name. Similarly, `line-too-long`

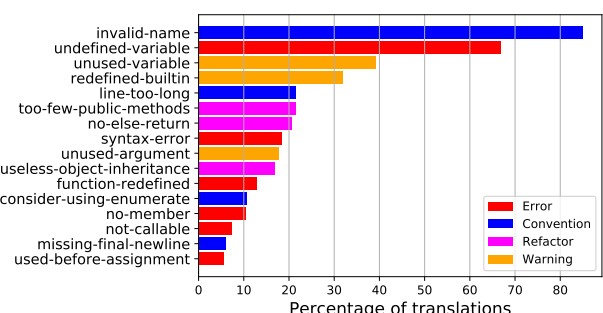

*Figure 1.* Lint violations and the percentage of translations in which they occur. We display violations that occur for more than 5% of the translations.

occurred in about 20% of the translations, where the model violated the recommended 80 character line length. Other violations that occurred frequently were `undefined-variable` (67%), `unused-variable` (39%), and `redefined-builtin` (32%); indicating that common programming conventions – to either not define an unused variable, or where built-ins could be overridden but were advised not to – were violated by the model.

These violations tie back to our user insight in Section 2.1.1, where participants expected the translated code to adhere to the conventions of the target language; and are in-line with the insights of Toral et al. (2018) in MT for natural languages, where the authors suggest document level translations to account for the intersentential phenomenon. Similarly, in the case of code translation, while function-level translation through TransCoder achieves around 60% accuracy (with a beam size of 5), translating whole classes requires models to account for auxiliary code and inter-function dependencies.

### 4.2. Quantitative Analysis

Another key insight that emerged from our user study was the need for interpretable model confidence values (see Section 2.1.2), to better help users focus on syntactical or conventional issues. To study the correlation between model confidences and lint violations through `Pylint`, we compute PBCC values for the two uncertainty metrics defined in Equations 1 and 2. Table 1 summarizes these results for the most commonly observed lint violations. We observe no correlation between model confidence: $p(y_t|y_{<t}, x, \theta)$, and tokens which resulted in lint violations. This miscalibration between model confidence and model interpretability has been studied in Guo et al. (2017), and was also pointed out by multiple users in our user study. In this work, we utilize only the decoder output probabilities to identify low confidence tokens, while Fomicheva et al. (2020) propose multiple metrics utilizing decoder output probabilities and attention weights for unsupervised quality estimation. Additionally, Guo et al. (2017) propose a calibration scheme to produce calibrated probabilities from deep neural networks.

Our results underline the need for further work on metrics that better align with human perception of code translation.

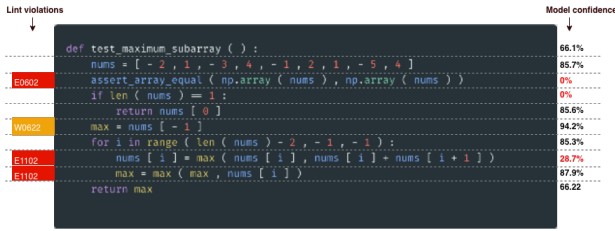

*Figure 2.* A code snippet translated from `Java` to `Python3` with the corresponding `Pylint` errors and model confidences for each line. The `max` variable is used by the TransCoder output both as a variable and as the `maximum` operator. As we frequently found, little correlation is visible between model confidences (right) and lint violations (left). See Appendix A.3 for the `Java` source code, and Figure 5 in the Appendix for an enlarged image.

### 4.3. Qualitative Example

We also present a specific translation instance along with lint violations and model confidences (Figure 2) to serve as an illustrative example of the nature of lint violations that occur in translated code, and how the model confidences values vary across the translation. While TransCoder correctly translates most of the code, including an interesting translation of `i>=0` in `Java` to `-1` in `Python3` in the `for` loop condition, it is unable to distinguish between the `Math.max` operator and `max` variable in `Java` – both of which have the same name but are different entities – and translates them over to the same variable but performing both functions in `Python3` (see violations `W0622` and `E1102` in Figure 2). The corresponding model confidences show some correlation with the lint violations with low confidences for `E0602` and `E1102` violations; but also show high confidences for `W0602` and the second `E1102` violation. This illustrates that model confidences do not correlate with associated programming errors or standards violations.

## 5. Conclusion & Future Work

In this work, we looked at automated code translation from the perspective of a programmer. From our user study, we found that the syntax and styling of the translation also matters to the users along with the code's executability and correctness; and an analysis of translations from the TransCoder model underscored the need for incorporating coding standards in the translation process. Additionally, users also desired some form of interpretability of the model's confidence in certain tokens. To quantitatively assess any correlation, we utilized the token probability values as a measure of the model's confidence, and lint violations as a surrogate metric for code interpretability We found no significant correlation. We are currently working on metrics that correlate better with a programmer's perception of code interpretability.

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

# A. Appendix

## A.1. User Study

See Figure 3 for the code translation design prototype.

*Figure 3.* Design prototype for a code translation user interface. Participants in the user study were shown this interface to demonstrate operation of the NMT model. *Left*: Input `Java` source code. *Right*: `Python` source code output, as translated by TransCoder (Roziere et al., 2020). Tokens in which the model was less than 95% confident are highlighted in red, with a tooltip displaying the model's confidence level.

## A.2. `Pylint` Details

1. `Pylint` URL: http://pylint.pycqa.org/en/latest/

2. For a full list of `Pylint` validations, please see: http://pylint-messages.wikidot.com/all-codes

3. Human readable messages for `Pylint` codes can be found at https://git.io/JTIma

4. Ignored `Pylint` checks:
   - `C0111` (missing-docstring): we ignore comments during the translation process
   - `C0326` (bad-whitespace): the detokenizer inserts spaces between all tokens
   - `R0201` (no-self-use): the dataset corresponds to algorithmic problems which do not require the use of `self` variable.

## A.3. Qualitative Example Source and Target Code

See Figure 4 for the qualitative example with the `Java` source and `Python` target code.

Refer to Figure 5 for an enlarged version of Figure 2.

```java
/**
 * Created by gouthamvidyapradhan on 07/07/2017. Find the contiguous
subarray within an array
 * (containing at least one number) which has the largest sum.
 *
 * <p>For example, given the array [-2,1,-3,4,-1,2,1,-5,4], the contiguous
subarray [4,-1,2,1] has
 * the largest sum = 6.
 */
public class MaximumSubarray {
  public static void main(String[] args) throws Exception {
    int[] nums = {-2, 1, -3, 4, -1, 2, 1, -5, 4};
    System.out.println(new MaximumSubarray().maxSubArray(nums));
  }

  public int maxSubArray(int[] nums) {
    if (nums.length == 1) return nums[0];
    int max = nums[nums.length - 1];
    for (int i = nums.length - 2; i >= 0; i--) {
      nums[i] = Math.max(nums[i], nums[i] + nums[i + 1]);
      max = Math.max(max, nums[i]);
    }
    return max;
  }
}
```

```python
def test_maximum_subarray ( ) :
    nums = [ - 2 , 1 , - 3 , 4 , - 1 , 2 , 1 , - 5 , 4 ]
    assert_array_equal ( np.array ( nums ) , np.array ( nums ) )
    if len ( nums ) == 1 :
        return nums [ 0 ]
    max = nums [ - 1 ]
    for i in range ( len ( nums ) - 2 , - 1 , - 1 ) :
        nums [ i ] = max ( nums [ i ] , nums [ i ] + nums [ i + 1 ] )
        max = max ( max , nums [ i ] )
    return max
```

Figure 4. Java source code (left) and the translated Python target code by TransCoder example used as a Qualitative Example in the Results section of the main paper.

Figure 5. A code snippet translated from `Java` to `Python3` with the corresponding `Pylint` errors and model confidences for each line. The `max` variable is used by the TransCoder output both as a variable and as the `maximum` operator. As we frequently found, little correlation is visible between model confidences (right) and lint violations (left). See Appendix A.3 for the `Java` source code.