# OpenReview forum: "Quality Estimation & Interpretability for Code Translation"
_NeurIPS.cc/2020/Workshop/CAP — NeurIPS 2020 CAP Workshop_

### Official Review · AnonReviewer1 · 2020-10-30
**Interesting insights into a deep learning model for code translation**

**Rating:** 7
**Confidence:** 4

**Review:**

This paper studies the quality of code translation using a blackbox deep neural network (DNN) model based on the same principles as DNNs used for machine translation. The key challenge is that evaluating the performance of these models in terms of the quality of the code generated (i.e., beyond correctness) is challenging.

To address this issue, they perform a user study to gain qualitative insights into the performance of the DNN, as well as a quantitative evaluation using a linter to diagnose both style issues as well as code errors. I thought the user study was particularly interesting, and the quantitative insights highlight some of the key challenges in applying machine learning to the programming domain (and possibly more broadly).

Their key findings include issues such as the presence of “obvious” mistakes, including high confidence ones. They also find that the DNN is not good at producing code in a certain style, though it might be possible to fine-tune it to do so.

Overall, I think the paper studies an important research problem and identifies some interesting insights.

---

### Decision · Program_Chairs · 2020-11-02

**Decision:**

Accept

**Comment:**

The review being positive, I am recommending acceptance.